# Bichir external gills arise via heterochronic shift that accelerates hyoid arch development

Jan Stundl[1,2], Anna Pospisilova[1], David Jandzik[1,3], Peter Fabian[1†],
Barbora Dobiasova[1‡], Martin Minarik[1§], Brian D Metscher[4], Vladimir Soukup[1]*,
Robert Cerny[1]*

[1]Department of Zoology, Faculty of Science, Charles University in Prague, Prague,
Czech Republic; [2]National Museum, Prague, Czech Republic; [3]Department of
Zoology, Faculty of Natural Sciences, Comenius University in Bratislava, Bratislava,
Slovakia; [4]Department of Theoretical Biology, University of Vienna, Vienna, Austria

*For correspondence:
vladimir.soukup@natur.cuni.cz
(VS);
robert.cerny@natur.cuni.cz (RC)

Present address: †Eli and Edythe
Broad CIRM Center for
Regenerative Medicine and Stem
Cell Research, University of
Southern California, Los Angeles,
United States; ‡The Prague
Zoological Garden, Prague,
Czech Republic; §Department of
Physiology, Development and
Neuroscience, University of
Cambridge, Cambridge, United
Kingdom

Competing interests: The
authors declare that no
competing interests exist.

Reviewing editor: Tanya T
Whitfield, University of Sheffield,
United Kingdom

**Abstract** In most vertebrates, pharyngeal arches form in a stereotypic anterior-to-posterior
progression. To gain insight into the mechanisms underlying evolutionary changes in pharyngeal
arch development, here we investigate embryos and larvae of bichirs. Bichirs represent the earliest
diverged living group of ray-finned fishes, and possess intriguing traits otherwise typical for lobe-
finned fishes such as ventral paired lungs and larval external gills. In bichir embryos, we find that
the anteroposterior way of formation of cranial segments is modified by the unique acceleration of
the entire hyoid arch segment, with earlier and orchestrated development of the endodermal,
mesodermal, and neural crest tissues. This major heterochronic shift in the anteroposterior
developmental sequence enables early appearance of the external gills that represent key
breathing organs of bichir free-living embryos and early larvae. Bichirs thus stay as unique models
for understanding developmental mechanisms facilitating increased breathing capacity.
DOI: https://doi.org/10.7554/eLife.43531.001

## Introduction

The vertebrate pharynx is composed of a series of repeated embryonic structures called pharyngeal
arches (*Graham, 2008*; *Grevellec and Tucker, 2010*). In the majority of jawed vertebrates, the first,
or mandibular arch contributes to the jaws; the second, or hyoid arch serves as the jaw support, and
the more posterior branchial arches typically bear internal pharyngeal gills. Pharyngeal arches form
in a highly stereotyped sequence from anterior to posterior, where the contacts between endoder-
mal pouches and surface ectoderm physically separate the mesoderm- and neural crest-derived arch
tissues (*Graham and Smith, 2001*; *Shone and Graham, 2014*; *Choe and Crump, 2015*). The pro-
gressive development of the pharynx has deep deuterostome origins, as it is characteristic of both
cephalochordates and hemichordates (*Willey, 1891*; *Gillis et al., 2012*; *Koop et al., 2014*). In verte-
brates, sequential formation of pharyngeal segments represents a fundamental aspect of the meta-
meric organization of the head and face (*Piotrowski and Nüsslein-Volhard, 2000*; *Couly et al.,
2002*; *Choe and Crump, 2015*). Any modifications of this well-established anteroposterior differenti-
ation scheme would represent a radical alteration in development of the stereotypic chordate bau-
plan (*Square et al., 2017*).

Polypterid bichirs represent the earliest diverged living group of ray-finned (Actinopterygian)
fishes (*Hughes et al., 2018*) and they are often referred to as the most relevant species for studying
character states at the dichotomy of ray- and lobe-finned fishes (e.g., *Standen et al., 2014*). This pla-
ces bichirs in a unique phylogenetic position among vertebrates, which can be exploited for

evolutionary and developmental comparative studies (e.g., *Takeuchi et al., 2009*; *Standen et al., 2014*; *Minarik et al., 2017*). Adult bichirs possess several intriguing characteristics that have been associated with air-breathing during the transition from water to land, such as ventral paired lungs or spiracular openings on the head (*Clack, 2007*; *Coates and Clack, 1991*; *Graham et al., 2014*; *Tatsumi et al., 2016*). Moreover, bichirs also share several key larval features with lungfishes or amphibians, such as cranial adhesive organs, and larval external gills (*Kerr, 1907*; *Diedhiou and Bartsch, 2009*).

The external gills of bichirs represent prominent adaptive structures, and constitute major breathing organs of their free-living embryos and early larvae (*Figure 1A*) (*Kerr, 1907*; *Diedhiou and Bartsch, 2009*). Strikingly, while external gills of amphibians and lungfishes derive from branchial arches as a rule (*Duellman and Trueb, 1994*; *Witzmann, 2004*; *Nokhbatolfoghahai and Downie, 2008*; *Schoch and Witzmann, 2011*), those of bichirs have historically been considered as unique hyoid arch derivatives due to their blood supply from the hyoid aortic arch (*Kerr, 1907*; *Goodrich, 1909*). Importantly, the external gills of bichir embryos represent the first cranial structures to appear, emerging before the eyes or mouth are evident (*Figure 1B*) (*Minarik et al., 2017*).

Here, we take advantage of an exceptionally complete embryonic series of the Senegal bichir (*Polypterus senegalus*) to explore the developmental underpinnings of the early formation of their external gills and test their segmental origin. Our results reveal that bichir external gills are definitively derived from the hyoid arch and develop by orchestrated acceleration of tissues of all germ layers of the hyoid segment. Thus, in bichir embryos, the standard anteroposterior differentiation scheme of cranial segments is modified by the unique heterochronic development of the hyoid metamere, allowing early and enhanced development of their external gills.

## Results and discussion

### External gills of the Senegal bichir are developmentally associated with the hyoid segment

In order to examine the origin of bichir external gills, we first followed the morphological development of this structure from the earliest stages of embryogenesis onwards. The first sign of external gill development is a pair of outgrowths situated lateral to the closing neural folds (*Figure 1C*). The hyoid origin of these outgrowths is suggested by the expression pattern of the *Hoxa2* (*Figure 1D*), a selector gene characteristic of hyoid identity in other vertebrates (*Rijli et al., 1993*; *Hunter and Prince, 2002*; *Baltzinger et al., 2005*). Later, at early pharyngula stages, the hyoid outgrowths produce protuberant bulges situated in the pre-otic region on each side of the embryo (*Figure 1E–H*), that rapidly increase in size (*Figure 1I*), and finally, differentiate into many secondary branches (*Figure 1J–L*). This suggests that the prominent external gills of bichir larva (*Figure 1A*) initially arise from striking accelerated development of the epidermal outgrowths (*Figure 1B*) that are of hyoid segmental origin (*Figure 1F*).

### Accelerated and predominant hyoid neural crest stream supplies bichir external gills

To gain insights into the accelerated development of the hyoid segment, we focused on the cranial neural crest that arises from the closing neural folds. Cranial neural crest cells emerge in a characteristic pattern and split into mandibular, hyoid, and branchial streams, which in most vertebrates arise in a sequential anteroposterior order of appearance. As a marker for migrating neural crest cells, we used expression of *Sox9*, a transcription factor critical for their emergence, migration, and differentiation (*Cheung and Briscoe, 2003*; *Mori-Akiyama et al., 2003*; *Theveneau and Mayor, 2012*). In bichir embryos, *Sox9* expression pattern reveals that the hyoid neural crest segment is developmentally advanced, as it forms concurrently with the mandibular neural crest segment (*Figure 2A*). Sections through the neural folds, however, demonstrate that mandibular neural crest cells still reside within the neuroepithelium (*Figure 2B*), while the hyoid neural crest cells have already emigrated from the neural folds (*Figure 2C*). This premature emigration of the hyoid neural crest stream correlates with the previously observed external outgrowths of the hyoid area (*Figure 1C*). Later in migration, the hyoid neural crest stream remains predominant (*Figure 2D*), as it is much larger when compared to the mandibular neural crest stream (*Figure 2E,F*). The hyoid neural crest stream still

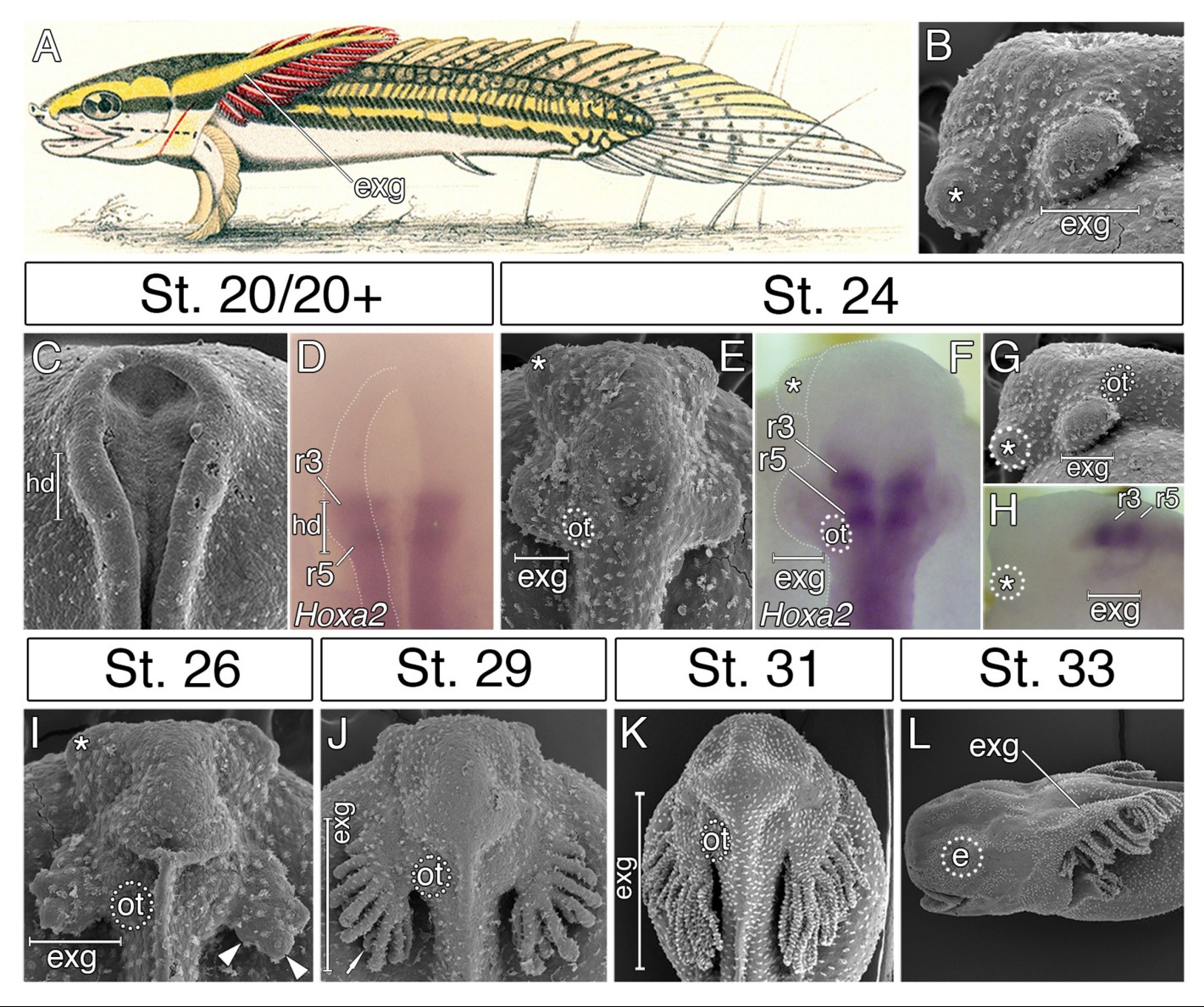

**Figure 1.** External gills of the Senegal bichir derive from the accelerated epidermal outgrowth of the hyoid segmental origin. (**A**) Budgett's illustration (**Kerr, 1907**) of a 3 cm long bichir larva with prominent external gills (exg). (**B**) Lateral view of an early pharyngula stage, SEM image showing external gills and cement glands (asterisk) as the first forming cranial structures. (**C**) SEM image of an early neurula stage with emerging bulge within the hyoid domain (hd). (**D**) *Hoxa2* expression in the neural tube at the level of the presumptive hyoid arch. (**E, G**) SEM images of a tailbud embryo with external gills anlage. (**F, H**) *Hoxa2* expression pattern in a tailbud stage, with highlighted position of external gills. (**I–L**) SEM images showing developmental morphogenesis of external gills. (**C–F, I–K**) Dorsal view. (**G–H, L**) Lateral view. e, eye primordium; ot, otic vesicle; r3, rhombomere 3; r5, rhombomere 5.
DOI: https://doi.org/10.7554/eLife.43531.002

progresses at later stages (*Figure 2G*), and as such, the majority of the mesenchyme in the early bichir head appears to arise from this source (*Figure 2H*). The Sox9 immunoreactivity further shows that cells of the leading edge of the hyoid stream delaminate from the neural folds prior to the emigration of the mandibular stream (*Figure 2I*), and illustrates the voluminous (*Figure 2J*) and extended (*Figure 2K*) mesenchymal production of the hyoid neural crest segment.

We directly tested whether the hyoid neural crest cell stream contributes to the external gills by performing focal CM-DiI injections into rhombomere 4 (*Figure 2L* inset), the source of the prospective hyoid neural crest stream in other vertebrates (*Lumsden et al., 1991*; *Köntges and Lumsden, 1996*; *Minoux and Rijli, 2010*; *Theveneau and Mayor, 2012*). One day after neurulation, the CM-

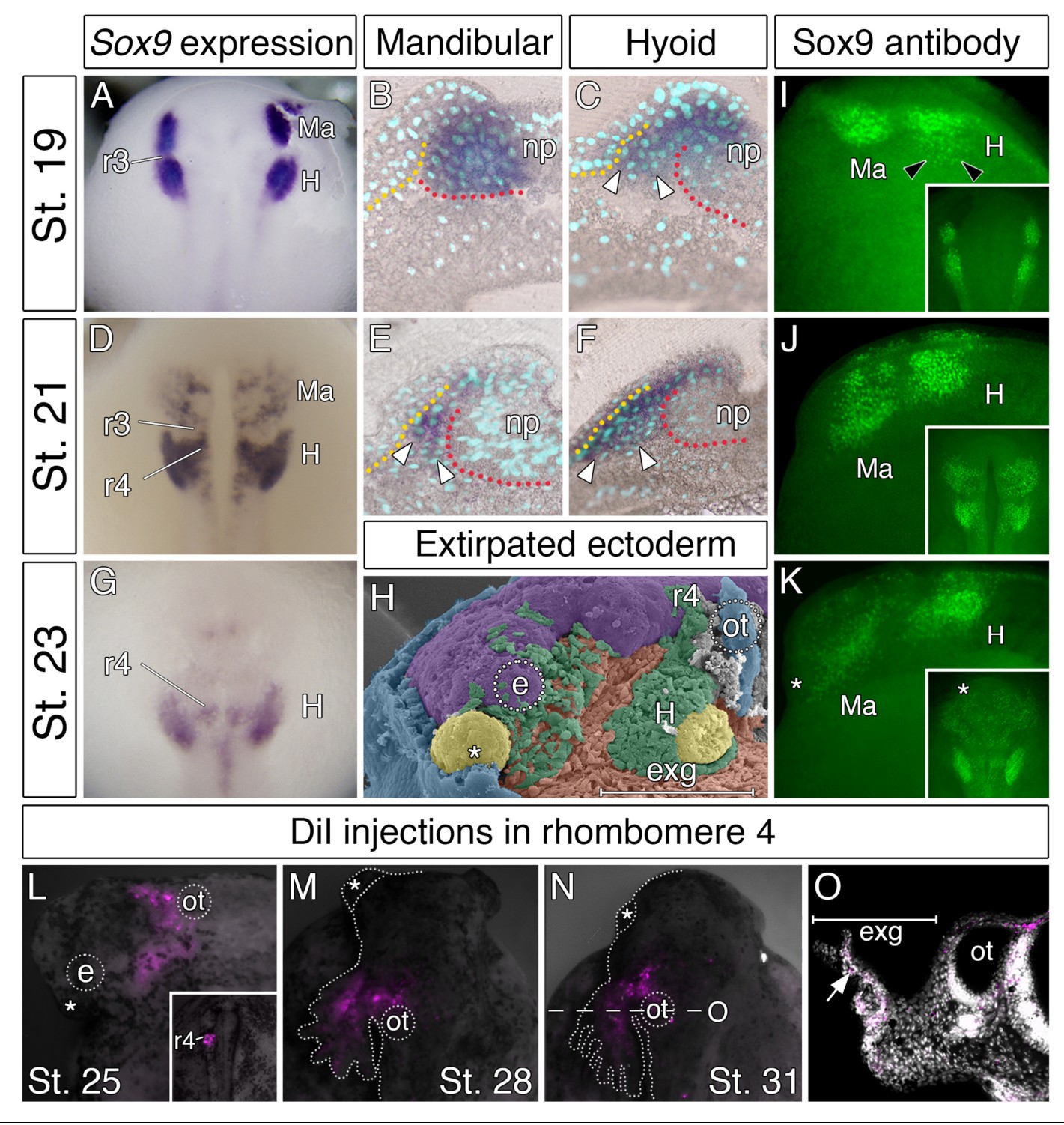

**Figure 2.** Accelerated formation and heterochronic development of the hyoid neural crest cells supply mesenchyme for the bichir external gills. (A, D, G) *Sox9* expression pattern in NC cells, from neurulation until early tailbud stages, dorsal views. Notice that the population of hyoid NC cells (marks as H) forms very early, and it later represents the most prominent cranial NC stream. (B–C, E–F) *Sox9* expression pattern in the mandibular and the hyoid domain, respectively, transversal sections. White arrowheads mark the ventral position of the NC cells. Dotted lines represent boundaries of neural- (red) and non-neural (yellow) ectoderm. DAPI (blue) shows cell nuclei. (H) Pseudocolored SEM image, lateral view on an embryo with the partially removed surface ectoderm (blue). NC cells are green, notice the amount of hyoid NC cells. Mesodermal mesenchyme is reddish, endodermal pouches are yellow, and the neural tube is violet. (I–K) Sox9 antibody visualizes individual neural crest cells. Lateral views, with small insets representing dorsal

*Figure 2 continued on next page*

*Figure 2 continued*

views. Black arrowheads in I show the advanced position of the hyoid NC cells. (L–O) Hyoid NC cell fate mapping (DiI red). Superimposed fluorescent and dark-field images at successive stages of development. (L) Lateral view, stage 25 embryo showing the hyoid NC stream. Small inset (dorsal view) represents an embryo at stage 20 immediately after the focal DiI injection into the rhombomere 4 (r4). (M–N) DiI signal at developing external gills, dorsal views. (O) Transversal section through the external gill (exg) at the level indicated in O. White arrow shows DiI signal in the primary branch of the external gill. Asterisk, cement gland; e, eye primordium; H, hyoid NC stream; Ma, mandibular NC stream; np, neural plate; ot, otic vesicle; r3, rhombomere 3; r4, rhombomere 4.

DOI: https://doi.org/10.7554/eLife.43531.003

DiI-positive hyoid neural crest cells are observed all along the proximodistal axis of the external gill primordium (*Figure 2L*). Two days later, they occupy the primary branches of the external gills (16/21, *Figure 2M*). After hatching, the CM-DiI-positive cells populate the fully developed and functional external gills (*Figure 2N,O*). Thus, our fate mapping experiment confirms that bichir external gills are, indeed, populated by the cells of the hyoid neural crest stream and, implicitly, that they represent hyoid arch derivatives.

## The first cranial muscles of bichir embryos support their external gills and are of hyoid segmental origin

In vertebrates, cranial neural crest cells are the primary source of craniofacial mesenchyme, but also have a major influence on the differentiation and morphogenesis of the cranial myogenic mesoderm (*Ericsson et al., 2004*; *Tokita and Schneider, 2009*). We, therefore, hypothesized that the pattern of cranial muscle differentiation in bichir embryos may be affected by acceleration of the hyoid neural crest segment (*Figure 2*). Whole-mount antibody staining against skeletal muscle marker 12/101 revealed that the first muscles differentiate stereotypically from the post-otic somites in the trunk region, as in other vertebrates (*Figure 3A*). However, within the cranial region of bichir embryos, the earliest developing muscles form within the hyoid arch and are associated with the external gills (*Figure 3B,C*). This first muscle complex (*levator and depressor branchiarum*, *Noda et al., 2017*) is situated lateral to the otic vesicle and connects filaments of the external gills to the gill stem (*Figure 3B–D*). The premature differentiation of the external gill-associated muscles is further supported by their innervation from the hyo-opercular ramus of the facial nerve, allowing voluntary movement of external gills from the earliest larval stages (*Figure 3E,F*). Other cranial muscles fully differentiate only at later larval stages when the external gill muscle complex becomes supplemented by other muscles of hyoid and mandibular origins (*Figure 3G*). Thus, bichir embryos display unique heterochrony in the differentiation of the hyoid over the mandibular arch mesoderm, providing muscular supports for their external gills.

## Early expansion of the hyoid endoderm triggers the formation of bichir external gills

Interestingly, the accelerated development of the external gill rudiments is also reinforced by the morphogenesis of the hyoid pharyngeal segment (*Figure 4A–J*). Reconstruction of the endodermal epithelium of the bichir pharynx using micro-CT imaging (*Minarik et al., 2017*) reveals that the pharyngeal endoderm forms two pairs of early outpocketings (*Figure 4B*). Whereas the rostral pair represents the embryonic precursor of the cement glands (*Figure 4A–D,F–I*) (*Minarik et al., 2017*), the posterior paired outpocketings constitute primordia of the external gills (*Figure 4A–D*). These posterior outpocketings belong to the hyoid segment, as the first pharyngeal pouch (mandibulo-hyoid, or spiracular) is situated rostrally (*Figure 4C,D*, white arrowhead) and the second pharyngeal pouch (hyoid-branchial) more caudally (*Figure 4H*, black arrowhead). Transverse sections confirm that these hyoid endodermal outpocketings constitute a substantial proportion of the external gill primordium (*Figure 4E*). At later stages, these outpocketings further transform into pocket-like structures (*Figure 4G,H,J*) that become supplemented with mesenchymal cells of the hyoid neural crest stream (*Figure 2L–N*). Thus, while ectoderm covers the entire external gill primordium, the endodermal outpocketing constitutes a considerable portion of the developing external gill (*Figure 4E*).

We sought to explore a possible role of the hyoid endodermal outpocketings in controlling development and morphogenesis of the bichir external gills. Morphogenesis of the pharyngeal pouches is critically regulated by factors from many signaling pathways (*Graham and Smith, 2001*;

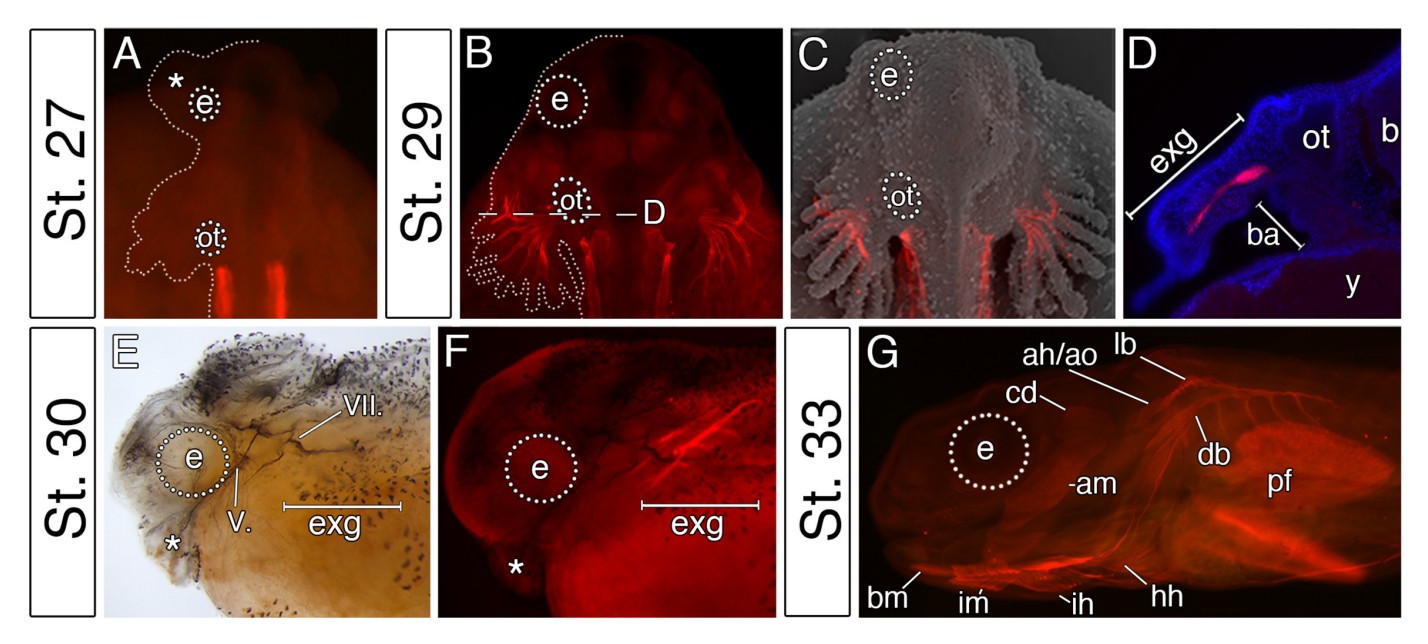

**Figure 3.** The premature differentiation of the external gill-associated cranial muscle complex in the Senegal bichir larva. (**A–C**) Dorsal view on bichir embryos, developing skeletal muscles are revealed by 12/101 antibody (red). The red signal in A (st. 27) refers to the post-otic somites. The first cranial muscle is associated with the external gills (B, stage 29). (**C**) Superimposed fluorescent and SEM image showing the context of the external gill muscles. (**D**) Transversal section through the external gills at the level indicated in B. DAPI (blue) stains cell nuclei. (**E, F**) Stage 30 bichir embryo, lateral view with (**E**) cranial nerves fibres labeled with anti-acetylated tubulin, and with (**F**) cranial muscles stained with 12/101 antibody (red). (**G**) Stage 33 bichir embryo, lateral view, with developed cranial muscles stained with 12/101 antibody (red). Asterisk, cement gland; am, adductor mandibulae; ah/ao, complex of adductor hyomandibulae and adductor operculi; b, brain; ba, branchial arches; bm, branchiomandibularis; cd,constrictor dorsalis; cement gland; e, eye primordium; lb/db, complex of levator branchiarum and depressor branchiarum; hh, hyohyoideus; ih, interhyoideus; im, intermandibularis; ot, otic vesicle; pf, pectoral fin; y, yolk; V., nervus trigeminus; VII., nervus facialis.

DOI: https://doi.org/10.7554/eLife.43531.004

*Graham, 2008*), among which alterations in Fibroblast growth factor (Fgf) signaling lead to defects in proper endodermal pouch development and pharyngeal segmentation (*Jandzik et al., 2014*; *Abu-Issa et al., 2002*; *Crump et al., 2004*; *Walshe and Mason, 2003*). To assess the possible role of Fgf signaling during bichir external gill development, we scored expression of the *Fgf8* ligand and the readouts of Fgf signaling activity. *Fgf8* expression is present in endodermal outpocketings and becomes confined to their lateral portions (*Figure 4—figure supplement 1*). These portions of endoderm in fact constitute the outgrowing tips of the prospective external gill (*Figure 4K*). Expression of *Dusp6* and *Pea3* (the Fgf signaling pathway readouts) and antibody localization for activated MAPK (marker of active Fgf signaling) are present in the external gill mesenchyme adjacent to the outgrowing endodermal tips or both in the mesenchyme and the endodermal tips (*Figure 4L–N*; *Figure 4—figure supplement 2*). The topographical relation of endodermal outpocketings and the direction of Fgf signaling within the external gill primordium thus suggest that the endodermal epithelium signals to the adjacent mesenchyme through Fgf signaling to regulate outgrowth of the external gill (*Figure 4O*).

To test the possible role of signaling events, we treated bichir embryos with SU5402, a collective Fgf and Egf signaling inhibitor, at early neurulation and scored the phenotypes at later pharyngula stages. In contrast to control embryos displaying well-developed hyoid endodermal outpocketings and external gill primordia (18/18, *Figure 4P–Q*), disrupting Fgf signaling perturbs morphogenesis of the hyoid endodermal outpocketings and leads to the loss of the external gill primordia (14/15, *Figure 4R–S*) possibly due to the loss of expression of downstream genes (*Figure 4T–U*). These results support a central role of the pharyngeal endoderm in triggering early development of bichir external gills. The pharyngeal origin of the external gill primordia is surprising given that the external gills are commonly considered as outer surface structures composed of ectoderm (*Takeuchi et al.,*

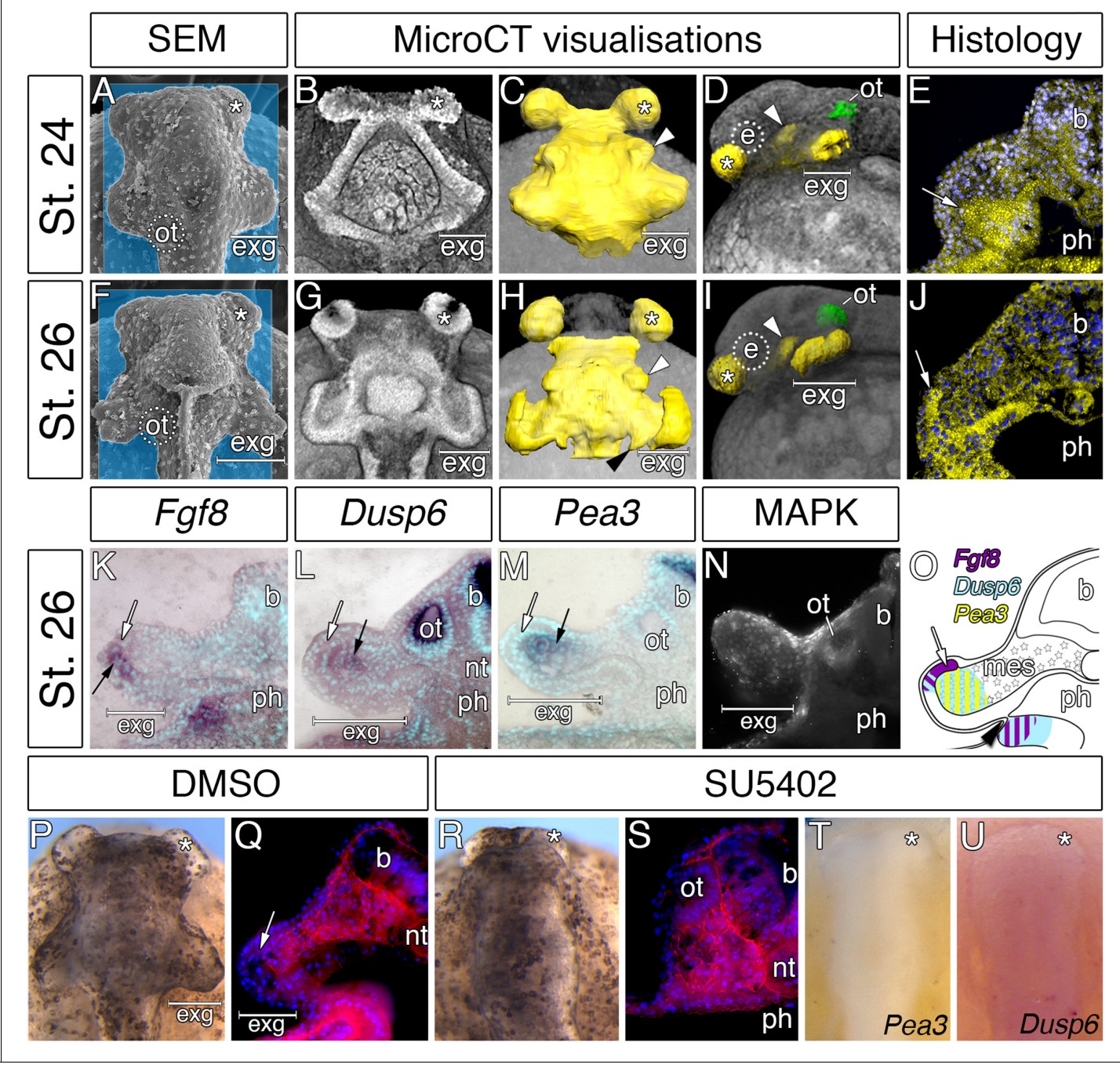

**Figure 4.** Considerable expansion of the hyoid pharyngeal endoderm contributes to the development of external gills in the Senegal bichir. (A, F) SEM images, dorsal view of bichir embryos with developing external gills (exg), showing the level of virtual sections in B and G. Notice the correspondence of the hyoid pharyngeal endoderm (B, G) and the external gills (A, F). (B–D, G–H) 3D models of pharyngeal endoderm (yellow) from dorsal (C, H), and lateral (D, I) view, respectively. (E, J) Transversal sections show prominent lateral expansion of hyoid pharyngeal endoderm (white arrow). (K–M) Transversal sections show wild-type expression of *Fgf8*, *Pea3*, and *Dusp6* (black arrow) in the external gills primordium. (N) Immunostaining of anti-activated MAP kinase antibody on transversal section of the external gills primordium. (O) Scheme summarizing *Fgf8*, *Dusp6*, and *Pea3* (K–M) expression patterns in the external gills formation at stage 26. Violet indicates *Fgf8* expression; blue marks *Dusp6* expression in the endoderm and adjacent mesenchyme of the external gills; yellow depicts expression of *Pea3* in the mesechyme of the external gills. (P–U) Inhibition of pouch-like endodermal outpocketings (P, R, T–U), dorsal view. (P–Q) Control larvae treated with DMSO develop normal pouch-like endodermal outgrowths (white arrow). (R) Larvae exposed to SU5402 from stage 20 till stage 26. (S) Transversal section shows loss of external gill anlagen. (T–U) SU5402 treated larvae fixed at stage 26 and probed for *Pea3* (T) and *Dusp6* (U). Nuclei are stained with DAPI (blue), basal laminae with anti-fibronectin (red). White

*Figure 4 continued on next page*

*Figure 4 continued*

arrowheads mark spiraculum (hyomandibular cleft) and black arrowhead marks hyo-branchial pouch. Asterisk, cement gland; b, brain; green, otic vesicle; e, eye primordium; nt, notochord; ot, otic vesicle; ph, pharynx.

DOI: https://doi.org/10.7554/eLife.43531.005

The following figure supplements are available for figure 4:

**Figure supplement 1.** Fgf8 expression during the course of bichir hyoid arch and external gill development.

DOI: https://doi.org/10.7554/eLife.43531.006

**Figure supplement 2.** Expression patterns of bichir Fgf8 *and transcriptional readouts of Fgf signaling,* Dusp6 *and* Pea3.

DOI: https://doi.org/10.7554/eLife.43531.007

---

*2009*; *Diedhiou and Bartsch, 2009*). However, our finding of an endodermal component in the early formation of bichir external gills reveals an unanticipated similarity with the true, internal gills of vertebrates, which typically form as pharyngeal endodermal structures (*Warga and Nüsslein-Volhard, 1999*; *Gillis and Tidswell, 2017*). Pharyngeal morphogenesis might thus represent a central developmental component of vertebrate gill breathing organs irrespective of their actual topographic position.

## Conclusions

The sequential formation of pharyngeal segments during embryonic development has deep deuterostome origins (*Willey, 1891*; *Koop et al., 2014*; *Gillis et al., 2012*) and it is well conserved among vertebrates, where all the embryonic cranial segments typically follow the sequential anteroposterior order during development (*Quinlan et al., 2004*; *Grevellec and Tucker, 2010*; *Schilling, 2008*; *Santagati and Rijli, 2003*). Bichir embryos diverge from this common scheme by the profoundly accelerated development of the second, hyoid segment, with earlier and orchestrated formation of the endodermal, mesodermal, and neural crest tissues (*Figure 5*). This unique heterochronic shift in the anteroposterior sequence constitutes a developmental basis for the early appearance of external gills that represent key breathing organs of bichir free-living embryos and early larvae.

Bichir external gills significantly differ from the external gills of amphibian and lungfish larvae that characteristically supplement the post-hyoid, branchial arches (*Duellman and Trueb, 1994*; *Witzmann, 2004*; *Nokhbatolfoghahai and Downie, 2008*; *Schoch and Witzmann, 2011*). The hyoid segmental origin represents a major developmental dissimilarity and suggests an independent evolution of bichir external gills. Developmentally, bichir external gills likely correspond to opercular structures that in ray-finned fishes typically form as caudal expansions of the hyoid arch to cover the gill-bearing branchial arches, and that persist in amniotes as early embryonic opercular flaps (*Richardson et al., 2012*). In bichirs, the opercular flap forms directly from the base of their external gills, and it progressively expands during early larval stages while external gills become reduced (*Diedhiou and Bartsch, 2009*). Interestingly, the hyoid arch-derived external gills and opercular flaps are both engaged in breathing and gill ventilation in bichir larvae. Moreover, in adult bichirs, the hyoid domain also contributes to air-breathing by forming paired spiracular chamber with openings located on the dorsal surface of the skull (*Graham et al., 2014*). Bichirs thus seem unique across recent vertebrates in enhancing breathing capacity through the development of several structures associated with the hyoid cranial segment.

## Materials and methods

### Embryo collection

Fish were manipulated in accordance with the institutional guidelines for the use of embryonic material and international animal welfare guidelines (Directive 2010/63/EU). Senegal bichir (*Polypterus senegalus* Cuvier, 1829) embryos were obtained, reared and staged as previously described (*Minarik et al., 2017*; *Diedhiou and Bartsch, 2009*). Embryos were dechorionated manually, fixed in 4% PFA in 0.1 M PBS at 4°C overnight, and then gradually dehydrated through a series of PBS/methanol mixtures and finally stored in 100% methanol.

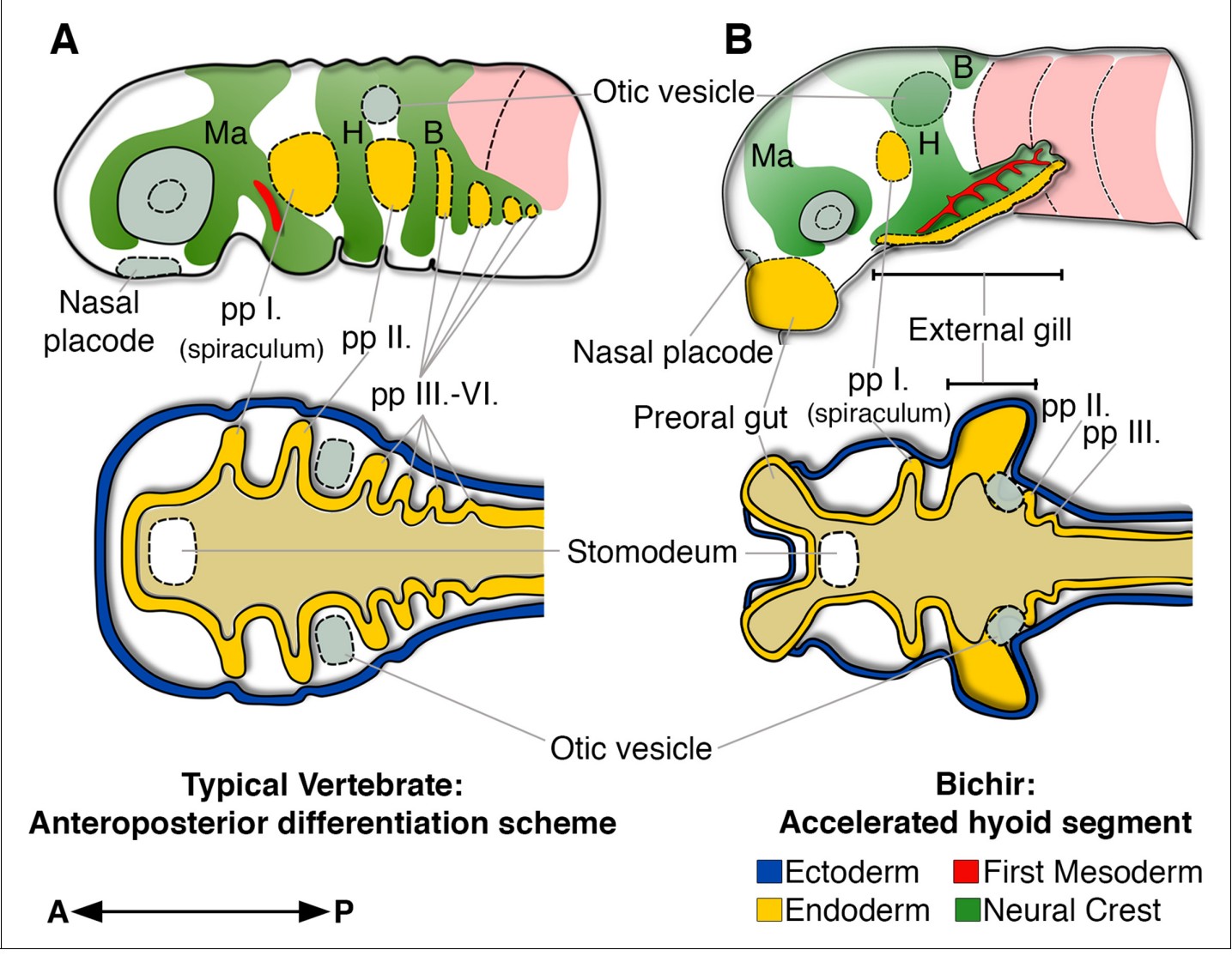

**Figure 5.** Bichir embryos diverge from the common anteroposterior differentiation scheme by accelerated development of the entire hyoid segment. (A, B) A cartoon of cranial neural crest migration (green), the first mesoderm (red), and pharyngeal pouches (yellow) in a typical vertebrate (A) and a bichir (B). Top are left lateral views, below are left horizontal sections. (A) In vertebrates, the sequential anteroposterior formation of cranial segments is well conserved, including pharyngeal pouches and cranial neural crest streams. (B) In bichirs, the entire hyoid segment is accelerated with earlier formation of the endodermal, mesodermal, and neural crest tissues, what constitutes a developmental basis for the appearance their external gills. Surface ectoderm in horizontal sections is shown in blue and primitive gut in ochre; B, branchial NC stream; H, hyoid NC stream; Ma, mandibular NC stream; pp I.-pp VI., pharyngeal pouches.

DOI: https://doi.org/10.7554/eLife.43531.008

### In situ hybridization and fate mapping

Whole-mount in situ hybridization with probes against *Hoxa2* (GenBank accession number: MK630352), *Sox9* (GenBank accession number: MK630350), *Fgf8* (GenBank accession number: MK630353), *Pea3* (GenBank accession number: MK630351), and *Dusp6* (GenBank accession number: MK630349) was performed as described (*Minarik et al., 2017*). Selected specimens were embedded in gelatine/albumin solution with glutaraldehyde, sectioned and counterstained with DAPI. Fate mapping experiments were carried out as described (*Minarik et al., 2017*). CM-DiI was injected into the neural fold of the prospective rhombomere 4 (*Figure 2L*). To confirm correct localisation of the tracking dye, some embryos were fixed immediately after injection, sectioned, and observed under

the fluorescent stereomicroscope in order to confirm proper localization of the cell tracking dye. The rest of the specimens were incubated until the desired stage and then fixed in 4% PFA in 0.1 M PBS.

### Scanning electron microscopy (SEM) and MicroCT imaging

Samples for SEM were fixed in modified Karnovsky's fixative (*Mitgutsch et al., 2008*). For direct visualization of cranial neural crest streams, the epidermis was removed using tungsten needles as described (*Cerny et al., 2004*). Specimens for MicroCT analysis were treated with phosphotungstic acid following the protocol developed by *Metscher (2009)* and scanned with a MicroXCT (X-radia) at the Department of Theoretical Biology, University of Vienna. Images were reconstructed in XMReconstructor (X-Radia), and virtual sections were analyzed in Amira (FEI Software).

### Antibody staining

Specimens for antibody staining were fixed in Dent's fixative. Muscles were labeled with 12/101 antibody (AB531892; Developmental Studies Hybridoma Bank), neural crest cells were labeled with Sox9 antibody (AB5535; Merck Millipore), basal lamina was labeled with anti-fibronectin (A0245; DAKO) and MAPK activity was assessed using anti-activated MAP kinase antibody (M8159; Sigma). Primary antibodies were detected by Alexa Fluor 488 and 594 (Invitrogen, Thermo Fisher Scientific Inc.). Visualisation of nerve fibres was performed using anti-acetylated tubulin antibody (T6793; Sigma) and EnzMet Enzyme Metallography kit (Nanoprobes).

### Pharmacological treatments

For inhibition of pharyngeal outpocketing morphogenesis, embryos were treated with 50 µM SU5402 in DMSO (Sigma Aldrich) from stage 20 until stage 26. Treatments were performed in E2 medium (*Brand et al., 2002*). Controls were reared in E2 medium with the equivalent DMSO concentrations.

## Acknowledgments

We thank Wojta Miller and Karel Kodejs for bichir colony care; James P. Cleland, Tatjana Haitina, Dan Medeiros, Rolf Ericsson and Jana Stundlova for critical reading of earlier versions of the manuscript; Martin Kralovic for initial work on this topic, Viktoria Psutkova and Kristyna Markova for technical assistance. This study was supported by the Charles University Grant Agency GAUK 1448514 (to JS), GAUK 640016 (to AP), GAUK 220213 and GAUK 726516 (to MM), the Charles University grant SVV 260434/2019 (to JS, AP, VS, DJ and RC), the Charles University Research Centre program No. 204069 (to VS), the grant of the Scientific Grant Agency of Slovak Republic VEGA 1/0415/17 and the European Union's Horizon 2020 research and innovation program under the Marie Skłodowska-Curie grant agreement No 751066 (to DJ), and the Czech Science Foundation GACR 16–23836S (to RC). Computational resources were supplied by the Ministry of Education, Youth and Sports of the Czech Republic under the Projects CESNET (Project No. LM2015042) and CERIT-Scientific Cloud (Project No. LM2015085) provided within the program Projects of Large Research, Development and Innovations Infrastructures.

## Additional information

### Funding

| Funder | Grant reference number | Author |
| --- | --- | --- |
| Charles University Grant Agency | 1448514 | Jan Stundl |
| Charles University Grant Agency | 640016 | Anna Pospisilova |
| Charles University Grant Agency | 220213 | Martin Minarik |
| Czech Science Foundation | 16-23836S | Robert Cerny |

| Charles University Grant Agency | 726516 | Martin Minarik |
| Charles University | Grant SVV 260434/2019 | Jan Stundl<br>Anna Pospisilova<br>David Jandzik<br>Vladimir Soukup<br>Robert Cerny |
| Charles University | Research Centre program 204069 | Vladimir Soukup |
| Vedecká Grantová Agentúra MŠVVaŠ SR a SAV | 1/0415/17 | David Jandzik |
| H2020 Marie Skłodowska-Curie Actions | 751066 | David Jandzik |

The funders had no role in study design, data collection and interpretation, or the decision to submit the work for publication.

### Author contributions

Jan Stundl, Conceptualization, Data curation, Investigation, Writing—original draft, Writing—review and editing; Anna Pospisilova, Martin Minarik, Data curation, Formal analysis, Investigation; David Jandzik, Validation, Investigation, Methodology; Peter Fabian, Formal analysis, Investigation, Methodology; Barbora Dobiasova, Brian D Metscher, Data curation, Formal analysis; Vladimir Soukup, Methodology, Writing—original draft, Writing—review and editing; Robert Cerny, Conceptualization, Funding acquisition, Writing—original draft, Writing—review and editing

### Author ORCIDs

Jan Stundl (iD) http://orcid.org/0000-0002-3740-3378
Anna Pospisilova (iD) http://orcid.org/0000-0002-8252-0709
Peter Fabian (iD) http://orcid.org/0000-0002-1096-6875
Martin Minarik (iD) https://orcid.org/0000-0001-6660-0031
Brian D Metscher (iD) http://orcid.org/0000-0002-6514-4406
Vladimir Soukup (iD) http://orcid.org/0000-0002-1914-283X
Robert Cerny (iD) http://orcid.org/0000-0002-0022-0199

### Decision letter and Author response

Decision letter https://doi.org/10.7554/eLife.43531.011
Author response https://doi.org/10.7554/eLife.43531.012

## Additional files

### Supplementary files

• Transparent reporting form
DOI: https://doi.org/10.7554/eLife.43531.009

### Data availability

All data generated and analysed during this study are included in the manuscript and providing files. All sources are cited in the Methods chapter.

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
