## [Decision Letter]

Thank you for submitting your article "Bichir external gills arise via a heterochronic shift that accelerates hyoid arch development" for consideration by *eLife*. Your article has been reviewed by three peer reviewers, and the evaluation has been overseen by Tanya Whitfield as Reviewing Editor and Diethard Tautz as the Senior Editor. The following individuals involved in review of your submission have agreed to reveal their identity: Andrew Gillis (Reviewer #1); Per E Ahlberg (Reviewer #2).

The reviewers have discussed the reviews with one another and the Reviewing Editor has drafted this decision to help you prepare a revised submission.

Summary:

This is a very interesting investigation into pharyngeal arch development in bichir. Bichir have a number of unusual features, which makes their study of interest from an evo-devo point of view, and this paper follows from an earlier study published by the group in Nature last year. Here the development of the larval external gills is followed, confirming the second arch origin of these elements with elegant DiI tracing and highlighting a heterochronic shift in pharyngeal arch development.

Essential revisions:

1) Reviewers 1 and 3 comment on the interpretation of the SU5402 experiments to inhibit Fgf signalling. They suggest further in situ hybridisation experiments: as a minimum, please aim to examine the expression of at least FGF8 and a transcriptional readout of Fgf signalling (e.g. *pea3* or *dusp6*). The reviewers predict that you will either see signalling interactions between pharyngeal mesoderm and endoderm (similar to what has been described in zebrafish), or some kind of signalling at the distal tip of the outgrowing gills (similar to what we see in the external gills of chondrichthyans). Either could fit with the result of your inhibition experiment. If there is no evidence of Fgf signalling, then the conclusion would need to be adjusted.

The reviewers are sympathetic of the challenges of working in systems with limited embryonic material, but hope that sufficient stocks of fixed embryos will be available to do the ISH. Thus, please attend to these revisions if at all possible. In the event that it is not possible to complete these experiments, the interpretations must be toned down as suggested.

2) Other suggested revisions do not involve further experimental work, but concern interpretations or conclusions drawn from the data. Please address these points or provide reasons for rebuttal.

The full reviews are appended below for your information.

*Reviewer #1:*

This manuscript reports on a very interesting study. The external gills of bichirs have long been viewed as an oddball structure in the field of vertebrate comparative anatomy, and little is known about their development (owing to the difficulty of obtaining and working with bichir embryos). These authors have produced some really nice work that blends anatomy and development in the bichir, *Polypterus senegalus*, and for this I commend them. The figures are very nice and clear, and support the authors' observations. The authors demonstrate convincingly that the external gills of bichir are of hyoid arch origin (as opposed to the post-hyoid gill arch origin of external gills in other vertebrate lineages), and they show that the hyoid arch of bichir exhibits a heterochronic acceleration of development (with relatively early/enhanced NC, mesodermal and pharyngeal endodermal contributions).

One section that might need a bit of revision is the text describing the final experiment with SU5402. It has been well established that loss of Fgf signalling results in failed endodermal pouch formation in zebrafish. We now know from work in Gage Crump's lab (e.g. Choe and Crump, 2015) that *Fgf8* functions in conjunction with *Wnt11r* as a guidance cue to direct pharyngeal endodermal pouch morphogenesis. The dynamics of this signalling are well known, with an origin of the *Fgf8* signalling from pharyngeal mesoderm, and a response in the pharyngeal endoderm. At the same time, there are many examples of Fgf signalling-dependence of lateral outgrowth (fins, limbs, gill filaments, gill arch appendages, etc). Here, the authors show that bichir embryos exposed to the pan-FGFR inhibitor SU5402 fail to form external gills, and conclude from this observation that pharyngeal endoderm "triggers" the early development of bichirs external gills. However, in the absence of gene expression data showing direction of signalling (e.g. which tissue expresses a gene encoding an Fgf ligand, and which tissue expresses a transcriptional readout of Fgf signalling), it is difficult to identify particular tissues as "drivers" of an Fgf-dependent developmental process. Is the failure of external gill initiation due to a failure of mesodermal-endodermal signalling (equivalent to the signalling that drives zebrafish pharyngeal pouch formation)? Or could it be because of a failure of some other source of Fgf signalling which contributes to the proximo-distal expansion of the external gills?

If it is not possible to carry out additional ISH experiments because of unavailability of embryonic material, then perhaps the authors could adjust their interpretations slightly, to be a bit more conservative – i.e. these experiments point to a likely Fgf-dependence of external gill formation, which could have a basis in X, Y or Z.

*Reviewer #2:*

This is an interesting manuscript that deserves publication. I have no concerns regarding the experiments or the conclusions that the authors draw from them. There are however a couple of points that I would like to discuss in relation to the Introduction and Discussion.

Introduction:

The authors claim that polypterids "originated soon after the key divergence of ray- and lobe-finned fishes" and cite Giles et al., 2017 in support of this statement. However, the conclusions of Giles et al., 2017 do not actually support that claim: quite the contrary. It used to be thought that polypterids were a really deep branch among the actinopterygii, with only a few fossil forms like Cheirolepis falling below them in the actinopt stem group. Giles et al. showed that polypterids derive from a Mesozoic actinopt group called scanilepids, and that this places them much higher up in the tree. They are still the deepest living branch, but they fall only just below the sturgeons and paddlefishes, and a whole slew of Devonian actinopterygians such as Moythomasia and its relatives (essentially what we used to call 'palaeoniscids') are now relegated to the actinopterygian stem. Giles et al., 2017 estimate the age of the split between polypterids and other living actinopts to somewhere around the end of the Devonian – say 360 million years. But the actinopt-sarcopt split almost certainly occurred during the late Silurian, no later than 420 million years at the earliest. So the polypterid lineage originated at least 60 million years after the actinopt-sarcopt divergence, which isn't what I would call "soon". Of course it is still true that they are topologically the closest living branch to the divergence on the actinopt side, just as the coelacanths are the closest branch on the sarcopt side.

Conclusion:

I am dubious about the argument regarding spiracle size and the specific relevance of polypterids for understanding the developmental basis of increased air-breathing capacity. It is true that polypterids have nice big spiracles that they use for breathing air, but the authors offer no evidence that this is in any way linked with their precocious hyoid arch development. And they are not unique: large spiracles are also seen in rays, where they are used for inhaling water. Furthermore, while the spiracles appear to have been large in early tetrapods and elpistostegids (see Brazeau and Ahlberg, 2006), this is not linked to a large spiracular arch; the hyomandibula in particular actually becomes progressively smaller across the fish-tetrapod transition. So if the final claim of the manuscript, that "bichirs might serve as models to understand the developmental basis of increased air-breathing capacity during the fish-tetrapod transition from water to land", specifically references the accelerated hyoid development of *Polypterus*, then I have to disagree. I see no particular reason to believe that the large spiracles of elpistostegids and early tetrapods were produced by accelerated hyoid arch development; indeed, given the conservation of the 'normal' front-to-back sequence of pharyngeal arch development in both tetrapods and non-polypterid fishes, there is good reason to believe that they were not.

*Reviewer #3:*

That the second arch develops before the first, throws out the usual anterior posterior pattern of differentiation observed in other vertebrates. The early migration of second arch crest is clearly shown using *Sox9*, although the mandibular stream appears to catch up fairly quickly.

Whether *Sox9* comes on first in the hyoid arch could probably be shown using a stage slightly earlier than st19. I was not sure of what Figure 2B, C, E, F, added to the paper and would probably remove these panels.

The Introduction states that "as a rule" the external gills of amphibians and lungfish are from the branchial stream (i.e. arches behind the first two arches (this should be defined)). How good is this data? Is it possible that all external gills are hyoid derived? The reference stated is a general text book on amphibians rather than original research data. If this is the only reference to the arch origin of external gills in amphibians and lungfish then perhaps there is no difference.

The authors show bulging endoderm at the second arch and suggest that the endoderm forms a substantial portion of the external gill primordium. They should probably qualify in the text that the ectoderm covers the gills, as shown in their final summary diagram. The endodermal bulges are shown at stage 24, while the neural crest changes are shown at stage 19. Therefore while the endoderm outpocketings might "prefigure development of the external gills", the neural crest changes come first, and the hyoid is already enlarged by this point.

The endodermal contribution, particularly at later stages when the gills have formed, cannot really be assessed by microCT. The authors have previously used DiI labelling to determine tissue origin and this would appear a better method. For example by labelling the ectoderm, which should then not label the forming gills.

The last part of the paper brings in functional experiments to test the roles of Fgf signalling using SU5402. The rational is to follow the role of hyoid endoderm, however, Fgfs are so widespread in the cranial region the experiments are certainly not specific to loss of Fgfs in the endoderm. This is particularly true as the experiments start inhibiting Fgfs from neurula stages. For these experiments to make sense we need to see a timecourse of Fgf expression/activity from neurula onwards. For example, it would be important to look at a readout of Fgf signalling, such as *Pea3*, and get an idea of what Fgf ligands might be expressed in the pharyngeal endoderm. Likewise we need to have confirmation that the inhibitor reduced *Pea3* expression. SU5402 can also inhibit the EGF pathway, so this should be mentioned. I appreciate this would probably require some gene cloning as this is a non-model organism but it should be possible.

The conclusion, therefore, that the endoderm triggers early development therefore cannot be made based on these experiments.

SU5402 can also inhibit the EGF pathway, so this should be mentioned.

---

## [Author Response]

Essential revisions:1) Reviewers 1 and 3 comment on the interpretation of the SU5402 experiments to inhibit Fgf signalling. They suggest further in situ hybridisation experiments: as a minimum, please aim to examine the expression of at least FGF8 and a transcriptional readout of Fgf signalling (e.g. pea3 or dusp6). The reviewers predict that you will either see signalling interactions between pharyngeal mesoderm and endoderm (similar to what has been described in zebrafish), or some kind of signalling at the distal tip of the outgrowing gills (similar to what we see in the external gills of chondrichthyans). Either could fit with the result of your inhibition experiment. If there is no evidence of Fgf signalling, then the conclusion would need to be adjusted.The reviewers are sympathetic of the challenges of working in systems with limited embryonic material, but hope that sufficient stocks of fixed embryos will be available to do the ISH. Thus, please attend to these revisions if at all possible. In the event that it is not possible to complete these experiments, the interpretations must be toned down as suggested.

Thank you very much for this major summary comment. Importantly, for this revised submission, we were capable to produce additional data involving gene expression patterns of *Fgf8, Pea3,* and *Dusp6* at consecutive stages of development, as described below in detail. As such, we now provide more data in Figure 4, and also add two new supplementary figures. In sum, we argue for the direction of Fgf signaling from the tip of the endoderm outgrowth to the underlying mesenchyme of bichir external gills. Please see more details below.

2) Other suggested revisions do not involve further experimental work, but concern interpretations or conclusions drawn from the data. Please address these points or provide reasons for rebuttal.

Thank you very much. We hope to address all the specific points raised by the reviewers below. We have reinforced some conclusions, reformulated some of our interpretations, and generally improved our revised manuscript according to reviewer's suggestions.

The full reviews are appended below for your information.Reviewer #1:[…] One section that might need a bit of revision is the text describing the final experiment with SU5402. It has been well established that loss of Fgf signalling results in failed endodermal pouch formation in zebrafish. We now know from work in Gage Crump's lab (e.g. Choe and Crump, 2015) that Fgf8 functions in conjunction with Wnt11r as a guidance cue to direct pharyngeal endodermal pouch morphogenesis. The dynamics of this signalling are well known, with an origin of the Fgf8 signalling from pharyngeal mesoderm, and a response in the pharyngeal endoderm. At the same time, there are many examples of Fgf signalling-dependence of lateral outgrowth (fins, limbs, gill filaments, gill arch appendages, etc). Here, the authors show that bichir embryos exposed to the pan-FGFR inhibitor SU5402 fail to form external gills, and conclude from this observation that pharyngeal endoderm "triggers" the early development of bichirs external gills. However, in the absence of gene expression data showing direction of signalling (e.g. which tissue expresses a gene encoding an Fgf ligand, and which tissue expresses a transcriptional readout of Fgf signalling), it is difficult to identify particular tissues as "drivers" of an Fgf-dependent developmental process. Is the failure of external gill initiation due to a failure of mesodermal-endodermal signalling (equivalent to the signalling that drives zebrafish pharyngeal pouch formation)? Or could it be because of a failure of some other source of Fgf signalling which contributes to the proximo-distal expansion of the external gills?If it is not possible to carry out additional ISH experiments because of unavailability of embryonic material, then perhaps the authors could adjust their interpretations slightly, to be a bit more conservative – i.e. these experiments point to a likely Fgf-dependence of external gill formation, which could have a basis in X, Y or Z.

Thank you very much for this vital comment. For this revised submission, we performed additional whole-mount and section analyses of *Fgf8* expression patterns at consecutive bichir stages 23-27 (Figure 4—figure supplement 1), and also tested the presence of transcripts of Fgf pathway readouts *Pea3* and *Dusp6* (Figure 4—figure supplement 2). Interestingly, we found the presence of *Fgf8* transcripts solely in the pharyngeal endoderm, with no signs of mesodermal *Fgf8* expression, in contrast to the zebrafish data of Choe and Crump (2015). Importantly, readouts of the Fgf pathway *Pea3* and *Dusp6* were found in the adjacent mesenchyme, or in a colocalization with *Fgf8* expression pattern in the endoderm. Based on these expression patterns, we infer the direction of Fgf signaling from the hyoid pouch endoderm to the underlying external gill mesenchyme. For further information on the expression patterns, and on the SU5402 experiment, please see also our response to reviewer 3.

Reviewer #2:This is an interesting manuscript that deserves publication. I have no concerns regarding the experiments or the conclusions that the authors draw from them. There are however a couple of points that I would like to discuss in relation to the Introduction and Discussion.Introduction:The authors claim that polypterids "originated soon after the key divergence of ray- and lobe-finned fishes" and cite Giles et al., 2017 in support of this statement. However, the conclusions of Giles et al., 2017 do not actually support that claim: quite the contrary. It used to be thought that polypterids were a really deep branch among the actinopterygii, with only a few fossil forms like Cheirolepis falling below them in the actinopt stem group. Giles et al. showed that polypterids derive from a Mesozoic actinopt group called scanilepids, and that this places them much higher up in the tree. They are still the deepest living branch, but they fall only just below the sturgeons and paddlefishes, and a whole slew of Devonian actinopterygians such as Moythomasia and its relatives (essentially what we used to call 'palaeoniscids') are now relegated to the actinopterygian stem. Giles et al., 2017 estimate the age of the split between polypterids and other living actinopts to somewhere around the end of the Devonian – say 360 million years. But the actinopt-sarcopt split almost certainly occurred during the late Silurian, no later than 420 million years at the earliest. So the polypterid lineage originated at least 60 million years after the actinopt-sarcopt divergence, which isn't what I would call "soon". Of course it is still true that they are topologically the closest living branch to the divergence on the actinopt side, just as the coelacanths are the closest branch on the sarcopt side.

Thank you very much for this important comment. In the revised submission, the problematic part of the sentence has been rephrased as follows: 'Polypterid bichirs represent the earliest diverged living group of ray-finned (Actinopterygian) fishes (Hughes et al., 2018) and they are often referred to as the most relevant species for studying character states at the dichotomy of ray- and lobe-finned fishes (e.g., Standen et al., 2014).'

Conclusion:I am dubious about the argument regarding spiracle size and the specific relevance of polypterids for understanding the developmental basis of increased air-breathing capacity. It is true that polypterids have nice big spiracles that they use for breathing air, but the authors offer no evidence that this is in any way linked with their precocious hyoid arch development. And they are not unique: large spiracles are also seen in rays, where they are used for inhaling water. Furthermore, while the spiracles appear to have been large in early tetrapods and elpistostegids (see Brazeau and Ahlberg, 2006), this is not linked to a large spiracular arch; the hyomandibula in particular actually becomes progressively smaller across the fish-tetrapod transition. So if the final claim of the manuscript, that "bichirs might serve as models to understand the developmental basis of increased air-breathing capacity during the fish-tetrapod transition from water to land", specifically references the accelerated hyoid development of Polypterus, then I have to disagree. I see no particular reason to believe that the large spiracles of elpistostegids and early tetrapods were produced by accelerated hyoid arch development; indeed, given the conservation of the 'normal' front-to-back sequence of pharyngeal arch development in both tetrapods and non-polypterid fishes, there is good reason to believe that they were not.

Thank you very much for this important comment. It seems that our comparisons were too superficial, and following your advice, we found our arguments were rather ambiguous. In the revised submission, therefore, the problematic part has been removed and we instead strengthen our data part.

Reviewer #3:That the second arch develops before the first, throws out the usual anterior posterior pattern of differentiation observed in other vertebrates. The early migration of second arch crest is clearly shown using Sox9, although the mandibular stream appears to catch up fairly quickly.Whether Sox9 comes on first in the hyoid arch could probably be shown using a stage slightly earlier than st19.

Thank you for your comments. We agree that the addition of the slightly earlier stage than stage 19 would nicely complement the table. The expected result would most likely be the presence of both the mandibular and hyoid neural crest cells still within the neuroepithelium. We were, however, unable to collect embryos at this stage.

I was not sure of what Figure 2B, C, E, F, added to the paper and would probably remove these panels.

The mentioned images represent transversal sections directly demonstrating that cells of the hyoid neural crest stream emigrate from the neural folds (2C) prior to the cells of the mandibular neural crest stream that, at the same stage, still reside within the neuroepithelium (2B). To our eyes, these sections provide direct evidence that cannot be seen on whole mount images (2A and D).

The introduction states that "as a rule" the external gills of amphibians and lungfish are from the branchial stream (i.e. arches behind the first two arches (this should be defined)). How good is this data? Is it possible that all external gills are hyoid derived? The reference stated is a general text book on amphibians rather than original research data. If this is the only reference to the arch origin of external gills in amphibians and lungfish then perhaps there is no difference.

Thank you for your comment. We agree that a general textbook as the single reference citation is not a sufficient prime source for the described phenomenon. In the revised submission, we have added several references to original research articles (Schoch and Witzman, 2011; Witzmann, 2004; Nokhbatolfoghahai and Downie, 2008) in order to support the branchial arch origins of external gills of amphibian and lungfish larvae. We have also specified that the branchial arches in fact represent the post-hyoid arches.

The authors show bulging endoderm at the second arch and suggest that the endoderm forms a substantial portion of the external gill primordium. They should probably qualify in the text that the ectoderm covers the gills, as shown in their final summary diagram.

Thank you for your comment. In the revised submission, we added information on the ectodermal coverage of the external gill primordia at the end of the first paragraph of the 2.4 section.

The endodermal bulges are shown at stage 24, while the neural crest changes are shown at stage 19. Therefore while the endoderm outpocketings might "prefigure development of the external gills", the neural crest changes come first, and the hyoid is already enlarged by this point.

Thank you for your comment. We agree that neural crest cells emigrate prior to lateral outpouching of the hyoid arch endoderm, and this situation resembles other vertebrate embryos, where neural crest cells appear first and lateral endodermal pouches later (see e.g. Cerny et al., 2004 for neural crest migration and pharyngeal morphogenesis in axolotl). We therefore removed the statement of endodermal outpocketings prefiguring development of the external gill. On the other hand, endoderm turns out to be the likely source for signals regulating outgrowth of the external gill (see also below).

The endodermal contribution, particularly at later stages when the gills have formed, cannot really be assessed by microCT. The authors have previously used DiI labelling to determine tissue origin and this would appear a better method. For example by labelling the ectoderm, which should then not label the forming gills.

Thank you indeed for this insightful comment. According to our data, microCT imaging can be used to track the endodermal contribution reliably only until bichir stage 26. Therefore, we performed CM-DiI injections into the pharynx in order to directly test the endodermal contribution to the external gills. Unfortunately, our CM-DiI labelling did not seem to hit specifically the hyoid segment endoderm (*n=*4). In parallel, we also performed labelling of the superficial ectoderm by carboxyfluorescein (CFDA-SE). However, the dye became quickly washed out from the embryonic surface (*n*=10). Thus, until now, all our attempts failed to discern the exact germ layer origin of the adult external gill epithelia. Despite this deficiency, we feel that the remaining data still show interesting phenomena connected with the early formation of bichir external gills.

The last part of the paper brings in functional experiments to test the roles of Fgf signalling using SU5402. The rational is to follow the role of hyoid endoderm, however, Fgfs are so widespread in the cranial region the experiments are certainly not specific to loss of Fgfs in the endoderm. This is particularly true as the experiments start inhibiting Fgfs from neurula stages. For these experiments to make sense we need to see a timecourse of Fgf expression/activity from neurula onwards. For example, it would be important to look at a readout of Fgf signalling, such as Pea3, and get an idea of what Fgf ligands might be expressed in the pharyngeal endoderm. Likewise we need to have confirmation that the inhibitor reduced Pea3 expression. SU5402 can also inhibit the EGF pathway, so this should be mentioned. I appreciate this would probably require some gene cloning as this is a non-model organism but it should be possible.The conclusion, therefore, that the endoderm triggers early development therefore cannot be made based on these experiments.

Thank you for your comment. In the revised submission, we rephrased the part about Fgf signaling inhibition, and also updatedFigure 4 with the new set of data. We added new gene expression patterns for *Fgf8* and for *Dusp6* and *Pea3*. The *Fgf8* expression is observed at the distal outgrowth of the forming external gill (see Figure 4K), apparently in the endodermal part (compare to Figure 4J). Expression of *Pea3* is visible in the mesenchyme adjacent to the *Fgf8*-expressing endoderm and expression of *Dusp6* is present both in the endoderm and in the adjacent mesenchyme (Figure 4L-M). Since Fgf ligands activate the MAPK signaling pathway, we also analyzed staining with anti-activated MAPK antibody. The antibody staining shows the signal throughout the external gill (Figure 4N).

Moreover, we analyzed the SU5402-treated embryos for the presence of *Pea3* and *Dusp6* transcripts (Figure 4T-U). Upon the treatment, we detected complete loss of expression of these genes, thus supporting the role of Fgf signaling in the development of bichir external gills.

SU5402 can also inhibit the Egf pathway, so this should be mentioned.

Thank you for your comment. In the revised submission, we mentioned affection of Egf pathway by SU5402 as suggested.